# Preoperative Balloon-Occluded Transcatheter Arterial Chemoembolization Followed by Surgical Resection: Pathological Evaluation of Necrosis

**DOI:** 10.3390/diseases11040149

**Published:** 2023-10-24

**Authors:** Jihoon Kim, Dong Il Gwon, Yonghun Kim, Gun Ha Kim, Seong Ho Kim, Hee Ho Chu, Jin Hyoung Kim, Ji Hoon Shin, Gi-Young Ko, Hyun-Ki Yoon

**Affiliations:** 1Department of Radiology, Research Institute of Radiology, University of Ulsan College of Medicine, Asan Medical Center, Seoul 05502, Republic of Korea; kimjihoon298@gmail.com (J.K.); kyh129@gmail.com (Y.K.); idgunkim@gmail.com (G.H.K.); evazzang23@gmail.com (S.H.K.); chuzzang1224@gmail.com (H.H.C.); jhkimrad@amc.seoul.kr (J.H.K.); jhshin@amc.seoul.kr (J.H.S.); kogy@amc.seoul.kr (G.-Y.K.); hkyoon@amc.seoul.kr (H.-K.Y.); 2Ajou University Hospital, College of Medicine, Ajou University, Suwon 16499, Republic of Korea

**Keywords:** liver, hepatocellular carcinoma, chemoembolization, balloon occlusion, tumor necrosis

## Abstract

This study investigates the clinical and pathological outcomes of preoperative balloon-occluded transcatheter arterial chemoembolization (B-TACE) in patients with single hepatocellular carcinoma (HCC). The data are from 25 consecutive patients who underwent sequential treatment of subsegmental B-TACE and hepatic surgery for single HCC. Radiological and pathological evaluation of oily subsegmentectomy, defined as the iodized oil-laden necrotic area that includes the entire HCC and surrounding liver parenchyma, were performed. Subsegmental B-TACE was technically successful in all patients. The major and minor complication rates were 8% and 24%, respectively. On the first follow-up computed tomography (CT), oily subsegmentectomy was observed in 18 (72%) out of 25 patients. Apart from one patient showing a partial response, the remaining 24 (96%) patients showed a complete response. Pathological complete necrosis of the HCC was observed in 18 (72%) out of 25 patients with complete or extensive necrosis of the peritumoral liver parenchyma. The remaining seven patients without peritumoral parenchymal necrosis had extensive necrosis of the HCCs. In conclusion, preoperative B-TACE can be a safe and effective method for the treatment of single HCC and a good bridge treatment for subsequent surgical resection. In addition, oily subsegmentectomy itself on the CT can be a good predictor of pathological complete necrosis of the HCC. The findings obtained from this study would provide a potential role of B-TACE in the treatment strategy for single HCC.

## 1. Introduction

Transcatheter arterial chemoembolization (TACE) is a therapeutic option for managing intermediate hepatocellular carcinoma (HCC), as recommended by the Barcelona Clinic Liver Cancer (BCLC) guidelines [1,2]. While hepatic resection or percutaneous ablative therapy currently stands as the endorsed curative treatment for BCLC stage A, a global analysis of clinical practices has revealed that TACE is applied to patients with early-stage HCC when curative options are restricted due to inoperable tumor site, portal hypertension, previous hepatectomy, and significant comorbidities [3,4,5,6]. The outcomes following TACE have experienced gradual improvements, largely attributed to advancements in medical device technology and the integration of cone-beam computed tomography (CT) along with advanced software for precise targeting of lesions, allowing for a more refined and selective approach [7]. Despite its clinical use in patients with early-stage HCC, incomplete necrosis is present within the treated HCC. Up to 57–77% of lesions retain viable tumors after TACE and thus require additional TACE or other treatment modalities [8,9,10,11].

Until recently, many studies have examined the effectiveness of preoperative TACE prior to surgery in patients with HCC [12,13,14,15]. Previous studies have also emphasized the importance of the degree of tumor necrosis attained through locoregional treatments [16,17,18,19,20]. Moreover, pathological complete necrosis to preoperative TACE was an independent factor for favorable overall survival after surgical resection and liver transplantation [21,22]. These results have indicated that HCC patients who showed pathological complete necrosis to bridging treatment, such as TACE, have a better prognosis after the operation.

To improve the therapeutic efficacy of TACE, balloon-occluded transarterial chemoembolization (B-TACE), in which a microballoon catheter is used for TACE, has been developed for HCC treatment. Temporary arterial occlusion by microballoon may improve treatment, possibly due to obstruction of proximal arterial flow that prevents reflux of chemotherapeutic and embolic agents [23,24,25]. Irie et al. reported that a decrease in balloon-occluded arterial stump pressure redistributes flow toward vascular regions with lower resistance, allowing the injection of chemotherapeutic and embolic agents under high pressure [23]. Although no definitive indications for B-TACE have been established to date, the procedure can increase the accumulation of iodized oil within the tumor to higher levels than occurs with conventional TACE (C-TACE), which improves local control in HCC [23,26]. Chu et al. suggested that B-TACE can be a better choice than C-TACE as an alternative treatment option in patients with single HCC [27]. However, no studies to date have evaluated the predictors and impacts of pathological complete necrosis following preoperative B-TACE in patients with single HCC. Therefore, the purpose of this study was to identify the clinical and pathological outcomes of preoperative B-TACE in patients with single HCC.

## 2. Materials and Methods

### 2.1. Study Population

Our institutional review board approved this single-center retrospective study and the requirement for obtaining written informed consent from the participants was waived. The records of our institution were retrospectively searched to identify patients who underwent sequential treatment with preoperative B-TACE and hepatic surgery for a single HCC between August 2018 and June 2022. The exclusion criteria were the intermediate, advanced, and terminal stages of the BCLC staging system (stage B/C/D). Patients were also excluded if they had undergone TACE previously or had a concurrent malignancy other than HCC. This study included 25 patients (19 men, 6 women; mean age, 59 years; range, 28–78 years). The baseline characteristics of the included patients are summarized in Table 1.

### 2.2. B-TACE

Preoperative B-TACE was undertaken to serve as a bridging treatment for patients with HCC. The B-TACE procedure was carried out by one of two interventional radiologists, each having respective experience of 6 and 20 years. After a right femoral artery puncture and cannulation, superior mesenteric and common hepatic angiography was performed using a 5 French catheter (specifically, the Rősch hepatic catheter by Cook, Bloomington, IN, USA). This procedure was conducted to evaluate several parameters, including the direction of portal blood flow, the anatomical characteristics of the hepatic artery, the presence of tumor vascularization, tumor location, and identification of feeding arteries. For selective B-TACE, a 2.0 F microballoon catheter (Optimo PB, Tokai Medical Products, Yokohama, Japan) was carefully advanced as closely as feasible to the target vessel. In cases where selective catheterization proved challenging due to vessel tortuosity or diminutive size, the microballoon catheter was positioned as near as possible to the artery supplying blood to the tumor. Following the correct positioning of the catheter, a microballoon was gently inflated to match the diameter of the target vessel. Subsequently, a standard protocol in our institution involved the routine administration of transarterial chemoinfusion before proceeding with the embolization process. In the majority of patients, a cisplatin-based TACE was carried out using a dose of 2 mg per kilogram of body weight. For patients who had allergies to cisplatin or exhibited renal dysfunction, doxorubicin-based TACE was employed, utilizing a dose ranging from 25 to 50 mg. An emulsion consisting of either cisplatin or doxorubicin and iodized oil (Lipiodol; manufactured by Guerbet, Roissy, France) in a 1:2 ratio was then injected. The endpoint of injection was the complete filling of iodized oil in the tumor and also extension into the intrahepatic collateral pathway and surrounding portal veins. Subsequently, a slurry of 100–300 µm gelatin particles (Nexsphere; Nextbiomedical, Incheon, Republic of Korea) was injected. The endpoint of embolization was densely filling the feeding artery with gelatin particles and stasis after balloon deflation. When a Gelfoam slurry beyond the catheter tip flowed into peripheral arteries, the slurry was promptly injected under microballoon deflation until it densely filled the arterial branches beyond the catheter tip.

Patients were closely observed in the ward for at least 3 days following B-TACE to evaluate their safety profile, including postembolization syndrome and other complications.

### 2.3. Radiological Evaluation and Portal Vein Embolization

Radiological evaluation was performed by analyzing the first post-B TACE follow-up CT scan according to the Modified Response Evaluation Criteria in Solid Tumors (mRE-CIST) criteria. This categorization system divides responses into four distinct groups: complete response (CR), partial response (PR), stable disease, and progressive disease [28]. Radiologic tumor response was categorized as objective response (CR and PR) or nonregression. At initial follow-up CT following B-TACE, iodized oil accumulation was quantitatively expressed in Hounsfield units (HU), which were measured by hand drawing a region of interest along the tumor margin on the axial image showing the maximum tumor diameter. These measurements were performed on pre-TACE CT images to provide a reference at a specific window level (50 HU) and width (350 HU). The presence of peritumoral parenchymal necrosis was also evaluated from the initial follow-up CT scan.

An indication for treatment by portal vein embolization (PVE) is less than 40% of the liver remaining in patients with liver cirrhosis. The PVE procedure has been described in previous studies [29,30]. Embolic materials, including the gelatin sponge particles (Gelfoam; Upjohn, Kalamazoo, MI, USA) and *n*-butyl cyanoacrylate (Braun AG, Melsungen, Germany), were used in combination with coils (Nester coil; Cook, Bloomington, IN) and/or an Amplatzer vascular plug (AGA Medical, Golden Valley, MN, USA).

### 2.4. Pathological Evaluation

Pathological evaluation of the surgical specimens was performed. In the gross examination, specimens were serially sectioned in 1 cm, and the diameter, location, and proportion of the solid components (possible viable tumor cells) of the tumor mass were examined. To evaluate viable tumor cells, at least 15 slices per case were examined, including every solid component cut for making formalin-fixed paraffin-embedded tissue, which was then cut into 4 µm sections and stained with hematoxylin and eosin (H&E). All slides were microscopically reviewed by two pathologists with at least 10 years of experience in hepatic pathology. HCC was diagnosed in specimens from all patients. The percentage of each specimen assessed to be tumor or necrotic surrounding parenchyma was recorded, and necrosis was classified as complete (100%), extensive (50–99%), or partial (<50%) by the pathologists as previously described [31].

### 2.5. Study Outcome Measurements and Statistical Analysis

Technical success after B-TACE was defined as the successful placement of a microballoon catheter into a tumor-feeding artery and transarterial injection of chemotherapeutic and embolic agents. Postembolization syndrome was defined as fever and/or nausea and/or pain (pain score >6 on a visual analog scale) presenting within 48 h following B-TACE [32]. Complications were categorized using the Society of Interventional Radiology clinical practice guidelines [33], which define major complications as admission to a hospital for therapy, an unplanned increase in the level of care, prolonged hospitalization (>48 h), permanent adverse sequelae, or death. Oily subsegmentectomy was defined as the iodized oil-laden necrosis area, including the HCC and peritumoral liver parenchyma of any thickness.

The Mann–Whitney U test was used to compare the median time interval between B-TACE and surgery in patients, whether they underwent PVE or not. Pre- and post-procedural serum alpha-fetoprotein and protein induced by vitamin K absence II (PIVKA-II) levels were compared using a Wilcoxon signed-rank test. Pre- and postprocedural serum parameters including serum aspartate transaminase (AST), alanine aminotransferase (ALT), and total bilirubin levels were also compared using the Wilcoxon signed-rank test. Statistical analyses were conducted using SPSS version 25 (IBM Corp. Chicago, IL, USA) and MedCalc Statistical Software version 20.1 (MedCalc Software, Ltd.). Statistical significance was set at *p* < 0.05.

## 3. Results

### 3.1. B-TACE Procedure

The procedure and treatment outcomes are presented in Figure 1. B-TACE was technically successful in all 25 patients (Figure 2). Cisplatin-based B-TACE was performed in 24 patients, and doxorubicin-based B-TACE was performed in the remaining patient.

Subsegmentectomy was defined as an iodized oil-laden necrosis area, including the HCC and peritumoral liver parenchyma. Extensive necrosis was defined as 50–99 percentage of necrosis of HCC quantitatively evaluated by pathologists.

The embolization covered one subsegment in twelve patients, two subsegments in ten, three subsegments in one, and four subsegments in two, respectively. Postembolic syndrome was experienced by 24% (6/25) of patients and is considered to be a minor complication. The rate of major complications after B-TACE was 8% (2/25). One patient developed ischemic biliopathy with infected biloma and was treated conservatively until liver resection surgery. Another patient developed prolonged fever (>7 days) and was treated with antibiotics and rehydration fluid for fever management, and the fever resolved within 12 days. Statistically significant elevation of laboratory data was observed for AST, ALT, and total bilirubin after B-TACE [AST: 33 ± 19 mmol/L vs. 361 ± 202 mmol/L (*p* < 0.001); ALT: 27 ± 12 mmol/L vs. 377 ± 241 mmol/L (*p* < 0.001); total bilirubin: 0.4 ± 0.5 mmol/L vs. 1.5 ± 0.7 mmol/L (*p* < 0.001)]. All parameters returned to the levels before B-TACE within 1 month (Figure 3).

PIVKA-II levels at the first follow-up visit (mean 20 days, range 15–31 days) were available in 24 patients, and both alphafetoprotein (median [IQR], 21 [4–405] to 5 [3–23]; *p* < 0.001) and PIVKA-II levels (median [IQR], 133 [46–511] to 29 [22–42]; *p* < 0.001) decreased significantly following B-TACE.

### 3.2. Radiological and Pathological Outcomes

The mean interval time between B-TACE and the first follow-up CT was 33 days (range 7–53 days), and the first follow-up CT from all patients was available for the study. According to the mRECIST criteria, twenty-four (96%) patients exhibited a CR, and one (4%) exhibited a PR. Therefore, all 25 patients achieved a radiological response on the first follow-up CT. The mean HU was 638 ± 187 (range, 215–967). Oily subsegmentectomy, including dense iodized oil-laden HCC and peritumoral liver parenchyma, was observed in CT images without contrast enhancement from 18 (72%) of 25 patients. The range of peritumoral parenchymal necrosis was confined within 4 cm from the HCC (Figure 4) in all 18 patients.

PVE was performed in 12 patients (48%). Among the 12 patients who underwent PVE, eight with oily subsegmentectomy had markedly attenuated peritumoral portal branches. Preoperative PVE was not significantly associated with the time interval between B-TACE and hepatic surgery (*p* = 0.854). The median interval between B-TACE and hepatic surgery was 39 days (95% CI, 31–47 days).

Pathology of 25 HCCs revealed complete necrosis in eighteen (72%) and extensive necrosis (mean, 96%; range, 90–99%) in seven (28%) (Figure 5). In the seven patients with extensive necrosis, 99% necrosis was observed in four, 98% in one, and 90% in two.

In the 24 HCCs classified as CR on follow-up CT pathological complete necrosis, pathological extensive necrosis was observed in eighteen and six HCCs, respectively. The accuracy of the mRECIST criteria was 76% (19/25), with a 25% (6/24) overestimation of tumor response on CT.

All 18 patients with oily subsegmentectomy were accompanied by pathological complete necrosis of HCCs. In these eighteen patients with oily subsegmentectomy, fourteen (77.8%) had pathological complete necrosis of peritumoral liver parenchyma, and four (22.2%) had pathological extensive necrosis (mean, 97%; range, 95–99%) (Table 2).

## 4. Discussion

Recently, it has been reported that HCC patients with underlying cirrhosis have better outcomes after liver surgery when they achieve pathological complete necrosis following preoperative TACE [21,22]. In the present study, preoperative B-TACE was successfully performed at the subsegmental level in all 25 patients. Pathological analysis showed that subsegmental B-TACE can be an effective treatment option for single HCC, achieving complete (100%) and extensive (mean, 96%; range, 90–99%) tumor necrosis in eighteen (72%) and seven (28%) patients, respectively. Several studies have conducted post-TACE pathological assessments on explanted liver specimens and have reported a range of complete tumor necrosis rates, varying from 16.7% to 44% [34,35,36,37]. Additionally, several prior studies have compared the effectiveness and safety of B-TACE and C-TACE in the treatment of HCC [38,39,40]. These studies have consistently shown that tumor response rates were notably higher in the B-TACE group in comparison to the C-TACE group. Control rates for primary HCCs at 1, 3, and 5 years were 92.4%, 69.9%, and 69.9%, respectively, in the B-TACE group, whereas they were 63.1%, 31.6%, and 25.3%, respectively, in the C-TACE group (*p* = 0.0016). In another study focusing on patients with intermediate-sized HCC (3–5 cm), it was found that the CR rate was significantly higher in the B-TACE group compared to the C-TACE group (72.3% vs. 54.1%; *p* = 0.047).

Previous studies suggested that the specific value of iodized oil accumulation was associated with treatment response after TACE [16,41,42,43], and that the degree of iodized oil accumulation is an important factor for predicting tumor response in post-TACE evaluation [31,44]. Park et al. [31] reported that the mean HU value of the iodized oil accumulation after C-TACE in tumors with pathological complete necrosis (547 ± 198 HU) was significantly higher than in tumors demonstrating extensive necrosis (369 ± 139 HU), and a threshold value for iodized oil accumulation >460 HU was highly sensitive and specific for pathologic complete necrosis. In the present study, the mean HU value of iodized oil accumulation was 638 ± 187 (range, 215–967) HU, which was higher than those of the previous studies predicting pathological necrosis. Therefore, in B-TACE, owing to the blocking effect of the proximal arterial flow, iodized oil can be forcefully infused into the tumor, resulting in dense iodized oil accumulation.

HCC often has satellite lesions that cannot be diagnosed by imaging modalities, and local tumor recurrences may occur when they go untreated. Previous histopathologic research reported that microsatellite lesions found in 46% of HCCs were smaller than 5 cm, and in 29% of HCCs, the lesions were smaller than 2.5 cm [45,46,47,48]. Therefore, in cases of HCC, it is necessary to treat not only the tumor itself but also the liver parenchyma around the tumor. However, previous studies only focused on the pathological necrosis of HCC itself [34,35,36,37]. To the best of our knowledge, this is the first study analyzing pathological necrosis of HCC and peritumoral liver parenchyma following subsegmental B-TACE. In the present study, we found that oily subsegmentectomy after B-TACE involving simultaneous necrosis of HCC and peritumoral liver parenchyma was achieved in 18 (72%) of 25 patients. Among the 18 patients with oily subsegmentectomy, all 18 also had complete necrosis of HCC. Moreover, regardless of the degree of pathological necrosis of peritumoral liver parenchyma, oily subsegmentectomy observed on CT can be a good predictor of pathologic complete necrosis of the HCC.

To induce a “subsegmentectomy-like” effect, the role of the microballoon catheter may be important because it can induce sufficient accumulation of iodized oil into the peritumoral portal veins through the peribiliary plexus and the drainage route from the HCC. Microballoon inflation leading to full embolization of the tumor-feeding artery and peritumoral portal veins can induce dense iodized oil accumulation in the tumor and extensive necrosis of peritumoral liver parenchyma, which ultimately lead to successful oily subsegmentectomy. Although we do not know the exact reason, peritumoral parenchymal necrosis was not achievable in all patients after successful subsegmental B-TACE. Moreover, the degree and range of peritumoral parenchyma necrosis cannot intentionally be controlled. However, oily subsegmentectomy was successfully achieved in 72% of patients and the range of peritumoral necrosis was always confined within 4 cm from the HCC. Our results thus do suggest that subsegmental B-TACE can induce dense iodized oil accumulation in the tumor and have sufficient embolic effects on peritumoral portal veins to induce peritumoral parenchymal necrosis. Therefore, if peritumoral parenchyma necrosis is identified on follow-up CT after subsegmental B-TACE, there would be no need for subsequent locoregional treatments because all the patients who had complete or extensive peritumoral parenchymal necrosis showed pathological complete necrosis of the HCC.

Another important aspect of the present study is related to safety. Several studies have reported that there is no significant difference in the rate of complications. The present study showed that the rate of major complications was 8%, which was comparable to that after C-TACE (5.6%) and similar to the rates in other studies [27,39,40,49]. In the present study, ischemic cholangiopathy was noted in only one patient. Another major complication was prolonged fever (>7 days). In addition, all patients experienced increased biochemical markers that included AST, ALT, and total bilirubin immediately after B-TACE, and 24% of patients experienced mild fever and/or nausea and/or pain defined as postembolization syndrome. However, the biochemical markers at the first follow-up had returned to the levels present before B-TACE (mean, 20 days; range, 15–31 days).

The absence of liver failure after treatment, and the self-resolving elevation in liver enzyme levels, is probably linked to the breakdown of tumor cells within the liver and could serve as a valuable indicator of a positive prognosis for predicting the response to treatment [50].

There are several limitations to the present study. First, there is the potential for selection bias because it was a retrospective single-institution study with a small study population. Second, the interval between B-TACE and surgery varied among the cases, and the pathologic evaluation of tumor response might vary according to the timing of the operation. Furthermore, the time interval between B-TACE and the subsequent image analysis was not consistent among patients, potentially leading to variations in the Hounsfield units (HU) of Lipiodol accumulation observed following B-TACE, depending on the timing of the initial CT scan. Nevertheless, the interval between B-TACE and the first CT or surgery was relatively brief for all patients, not exceeding a duration of 2 months. Third, we only included patients with single HCC, not those with intermediate HCC that generally warrant C-TACE. To date, there are no widely agreed indications for B-TACE, although previous studies reported that the multiplicity of HCC was an indicator of the ineffectiveness of B-TACE and suggested that the optimum indication of B-TACE was single HCC [49,51]. Finally, the additional influence of PVE on tumor necrosis is not considered. The effect of PVE on HCC remains controversial and is not well established. However, the effect of PVE would be minimal, especially in patients with oily subsegmentectomy due to markedly attenuated peritumoral portal veins. Nevertheless, a multicenter prospective randomized controlled trial is warranted.

## 5. Conclusions

Preoperative B-TACE can be a safe and effective method for the treatment of single HCC and a good bridge treatment for subsequent surgical resection. In addition, oily subsegmentectomy on CT scans can serve as a reliable predictor of complete pathological necrosis in HCC, making it an indicator of effective treatment. We hope findings from this study can enhance the potential role of B-TACE in the treatment strategy of single HCC.

## Figures and Tables

**Figure 1 diseases-11-00149-f001:**
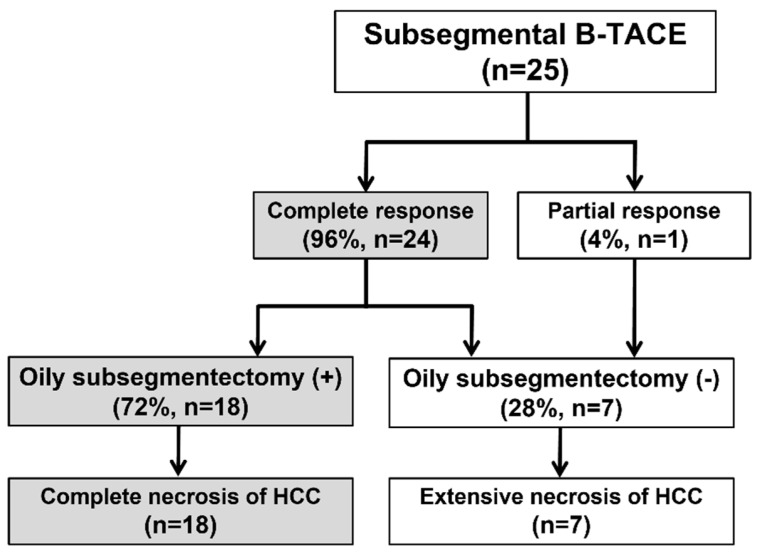
Flow chart of procedure and treatment outcome. B-TACE, balloon-occluded transcatheter arterial chemoembolization; HCC, hepatocellular carcinoma; Oily.

**Figure 2 diseases-11-00149-f002:**
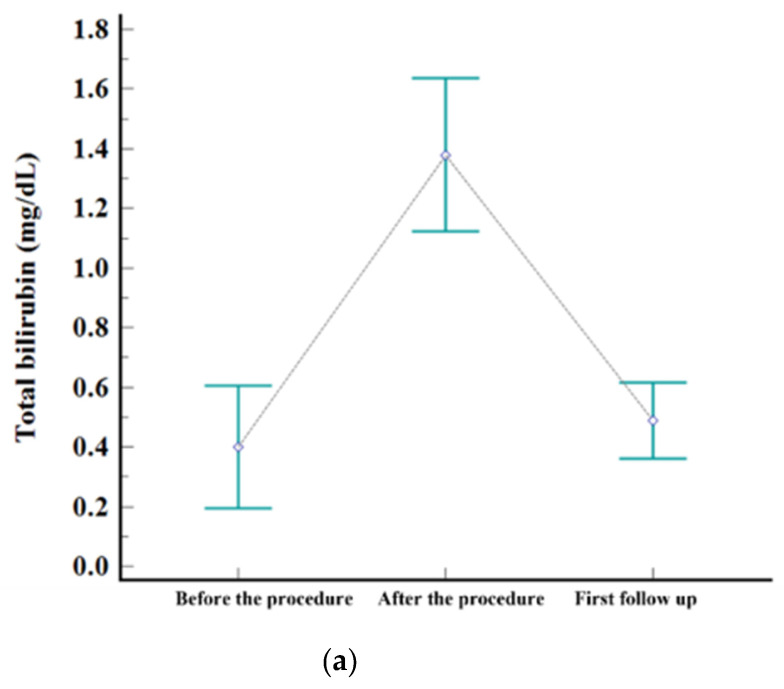
Levels of serum. (**a**) AST, (**b**) ALT, and (**c**) total bilirubin before the procedure, after the procedure (highest level during admission), and at first follow-up visit.

**Figure 3 diseases-11-00149-f003:**
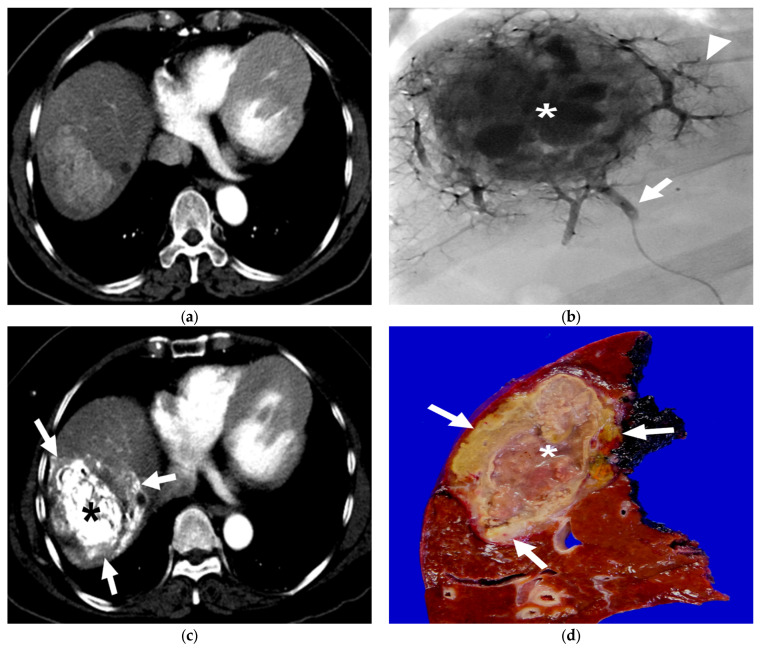
A 63-year-old female patient with single HCC (6.8 cm). She underwent B-TACE and sequential right hemihepatectomy after 43 days. (**a**) Preprocedural multiphase axial CT with arterial phase reveals an arterial enhancing lesion, indicating hypervascular HCC in liver segment 8. (**b**) A single fluoroscopy image during subsegmental B-TACE showing the HCC (asterisk) and peritumoral portal veins (arrowheads) markedly opacified with Lipiodol. Note the inflated microballoon (arrow) at the right anterior superior hepatic artery. (**c**) Arterial phase of dynamic CT image obtained 2 weeks after B-TACE shows successful oily subsegmentectomy including dense lipiodol accumulation in the HCC (asterisk) and peritumoral necrosis (arrows) with heterogenous lipiodol accumulation. Note the HU of HCC is 561. (**d**) Gross photograph of the resected specimen with totally necrotic mass. Note that well-defined brownish-yellow necrotic HCC (asterisk) and yellowish-white necrotic parenchyma (arrows).

**Figure 4 diseases-11-00149-f004:**
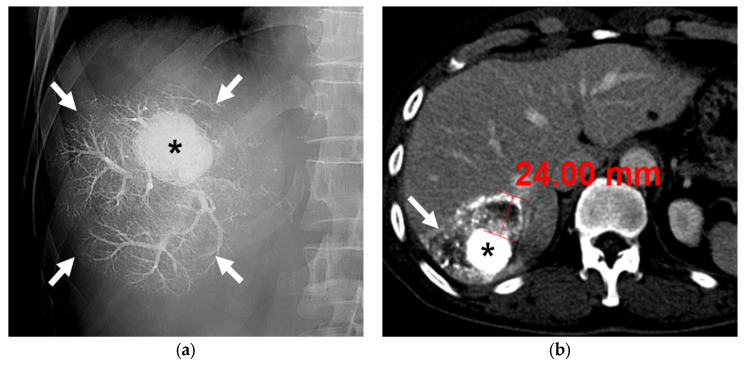
A 63-year-old male patient with single HCC (3.2 cm). (**a**) Fluoroscopic image after subsegmental B-TACE at the right posterior superior hepatic artery shows dense lipiodol accumulation in the HCC (asterisk) and peritumoral portal veins (arrows). (**b**) Portal phase CT image obtained 1 month after B-TACE shows complete oily subsegmentectomy including compact and dense lipiodol accumulation (HU: 902) in the HCC (asterisk) and peritumoral necrosis (arrows). Note the HU of the HCC is 902 and the maximum width of the parenchymal necrosis is 24 mm. Surgical specimen (not shown) showed complete necrosis of the HCC and peritumoral liver parenchyma.

**Figure 5 diseases-11-00149-f005:**
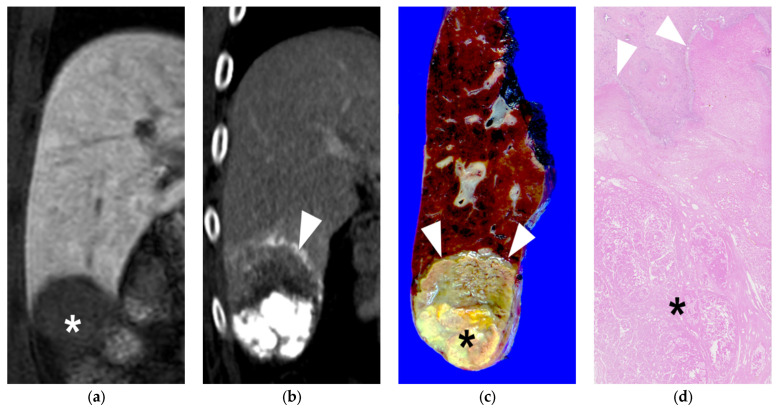
A 60-year-old male patient with single HCC (3.2 cm). He underwent B-TACE and sequential right posterior sectionectomy after 1 month. (**a**) 3D fat-suppressed spoiled gradient echo T1-weighted coronal image 20 min after contrast injection showed low signal intensity of HCC in liver segment 6 (asterisk). (**b**) 15 days f/u coronal CT image (portal phase) showed dense lipiodol accumulation (HU: 579) in HCC without viable portion, which was CR in the mRECIST criteria. Note that necrosis of the peritumoral parenchyma (arrowheads). (**c**) Gross photograph of the resected specimen with totally necrotic mass in liver segment 6. Note that well-defined brownish-yellow necrotic HCC (asterisk) and yellowish-white necrotic parenchyma (arrows). (**d**) Microscopic findings of the resected hepatic mass showing complete necrosis of HCC (asterisk) and peritumoral liver parenchyma (H&E stain, ×10).

**Table 1 diseases-11-00149-t001:** Characteristics of the Patients Included in the Study.

Characteristics (n = 25)	Value
Age (years)	59 ± 12 (28–78)
Sex	
Men	19 (76%)
Women	6 (24%)
Location of HCC	
Peripheral	21 (84%)
Central	4 (16%)
Lesion size (mm)	45 (20–100)
Portal vein embolization	12 (48%)
Cause of liver cirrhosis	
Hepatitis B virus infection	17 (68%)
Hepatitis C virus infection	2 (8%)
Alcohol	2 (8%)
Others	4 (16%)
Child-Pugh class	
A	24 (94%)
B	1 (6%)
Aspartate aminotransferase (U/L)	33 ± 19
Alanine aminotransferase (U/L)	27 ± 12
Total bilirubin (mg/dL)	0.4 ± 0.5
a-Fetoprotein (ng/mL), median (IQR)	21 (4–405)
PIVKA-II (mAU/mL), median (IQR) ^a^	133 (46–511)

^a^ Data available for 24 patients. HCC = hepatocellular carcinoma; IQR = interquartile range; PIVKA = protein induced by vitamin K absence or antagonist.

**Table 2 diseases-11-00149-t002:** Outcomes of Radiologic and Pathologic evaluation.

	Total Patients (n = 25)	Patients with Oily Subsegmentectomy (n = 18)
Radiologic evaluation		
Complete response	24 (96%)	18 (100%)
Partial response	1 (4%)	0 (0%)
Pathologic evaluation of HCC		
Complete necrosis	18 (72%)	18 (100%)
Extensive necrosis	7 (28%)	0 (0%)
Pathologic evaluation of peritumoral liver parenchyma		
Complete necrosis	14 (56%)	14 (78%)
Extensive necrosis	11 (44%	4 (22%)

## Data Availability

The data presented in this study are available upon justified request.

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
