# Peer review of "Preoperative Balloon-Occluded Transcatheter Arterial Chemoembolization Followed by Surgical Resection: Pathological Evaluation of Necrosis"

_diseases, 2023, doi:10.3390/diseases11040149_

Round 1
Reviewer 1 Report
1. General Comments:
The manuscript presents a study on the clinical and pathological outcomes of preoperative balloon-occluded transcatheter arterial chemoembolization (B-TACE) in patients with single hepatocellular carcinoma (HCC). The study is well-organized, and the results are systematically presented. The topic is of clinical importance, and the findings could have significant implications for the treatment of HCC.
2. Major Comments:
Abstract: The abstract provides a clear overview of the study's objectives, methods, and results. It would be beneficial to include a brief statement on the study's significance or implications for clinical practice.
Introduction: The introduction provides a comprehensive background on TACE and its role in HCC treatment. However, it would be beneficial to provide more context on the significance of preoperative B-TACE and its potential advantages over traditional TACE.
Materials and Methods: The methods section is detailed and provides a clear description of the study population, B-TACE procedure, radiological and pathological evaluations, and statistical analysis. However, it would be helpful to provide more details on the criteria used for patient selection and the rationale behind the chosen methods.
Results: The results section presents the outcomes of the B-TACE procedure, radiological and pathological evaluations, and the statistical significance of the findings. It would be beneficial to include more visual aids, such as tables or figures, to summarize the key results.
Discussion/Conclusion: The provided summary does not include a discussion or conclusion section. It is crucial to include a discussion section to interpret the results, compare them with existing literature, and provide potential implications and future directions.
3. Minor Comments:
Formatting: Ensure that the manuscript adheres to the journal's formatting guidelines, especially in the references and tables.
Language: The manuscript is generally well-written, but there are some minor grammatical errors that need correction.
4. Conclusion:
The study provides valuable insights into the clinical and pathological outcomes of preoperative B-TACE in patients with single HCC. The findings suggest that B-TACE can be a safe and effective method for treating single HCC and can serve as a bridge treatment for subsequent surgical resection. However, there are areas that need improvement, particularly in the presentation of results and the inclusion of a discussion section. Once these revisions are made, the manuscript will be a strong contribution to the field.
Author Response
Reviewer 1
- General Comments:
The manuscript presents a study on the clinical and pathological outcomes of preoperative balloon-occluded transcatheter arterial chemoembolization (B-TACE) in patients with single hepatocellular carcinoma (HCC). The study is well-organized, and the results are systematically presented. The topic is of clinical importance, and the findings could have significant implications for the treatment of HCC.
- Major Comments:
Abstract: The abstract provides a clear overview of the study's objectives, methods, and results. It would be beneficial to include a brief statement on the study's significance or implications for clinical practice.
# Thank you for your valuable comment. We completely agree with your opinion that adding a description regarding the clinical significance of B-TACE in the abstract would enhance the comprehensiveness of this study. Therefore, we added the sentence that clinical implication of this study in the abstract (lines 26-27).
Introduction: The introduction provides a comprehensive background on TACE and its role in HCC treatment. However, it would be beneficial to provide more context on the significance of preoperative B-TACE and its potential advantages over traditional TACE.
# Thank you for this important comment and helpful suggestions. To the best of our knowledge, there was no study about preoperative B-TACE. Our study is the first report about preoperative B-TACE. We agree with your opinion that providing a more description of the comparative advantages between B-TACE and traditional TACE would be helpful for emphasize the topic of this paper. So, we added some comments about the comparison between B-TACE and C-TACE in the Introduction section (lines 53-55).
Materials and Methods: The methods section is detailed and provides a clear description of the study population, B-TACE procedure, radiological and pathological evaluations, and statistical analysis. However, it would be helpful to provide more details on the criteria used for patient selection and the rationale behind the chosen methods.
# Thank you for this valuable comment. Following your recommendation, we have added more details about patient selection in the Materials and Methods (lines77-78).
Results: The results section presents the outcomes of the B-TACE procedure, radiological and pathological evaluations, and the statistical significance of the findings. It would be beneficial to include more visual aids, such as tables or figures, to summarize the key results.
# Thank you for this helpful suggestion. We agree with your opinion that this study need a table for summarizing the main results. Therefore, we have now added the summarized outcomes of the study to the new table (lines 261; Table 2).
Discussion/Conclusion: The provided summary does not include a discussion or conclusion section. It is crucial to include a discussion section to interpret the results, compare them with existing literature, and provide potential implications and future directions.
# Thank you for this valuable comment. According to your recommendation, we have added this point to the Discussion section (lines 324-327 and lines 354-356).
- Minor Comments:
Formatting: Ensure that the manuscript adheres to the journal's formatting guidelines, especially in the references and tables.
# We used the Endnote style recommended by the Diseases journal (MDPI ACS Journals) to write the references. We have made some adjustments to the text layout in Table 1. If there are any further modifications required, please let us know, and we will make the necessary additional revisions. Thank you for your comment.
Language: The manuscript is generally well-written, but there are some minor grammatical errors that need correction.
# We apologize for our mistakes. We have corrected this in the manuscript.
- Conclusion:
The study provides valuable insights into the clinical and pathological outcomes of preoperative B-TACE in patients with single HCC. The findings suggest that B-TACE can be a safe and effective method for treating single HCC and can serve as a bridge treatment for subsequent surgical resection. However, there are areas that need improvement, particularly in the presentation of results and the inclusion of a discussion section. Once these revisions are made, the manuscript will be a strong contribution to the field.

Reviewer 2 Report
1. Why the ratio of man and women are huge, write the reason.
2. For age, did you follow the medical worldwide rule mention, cite it in table 1.
3. Expand the abbreviation AST and ALT for more understanding when use for the first time
4. In introduction author should add the names of authors previous work in this field.
5. In Table 2, the words on x-axis are blurred make it clear for one eye.
6. Conclusion is very short, elaborate it with your best results in discussion.
7. The word “in conclusion” in abstract is not meaningful. Use word “we hope”.
8. Remove all type mistakes and space issue from the manuscript.
9. Ther are some complications linked with BTACE, did you check it mention it.
10. Before starting BTACE, give one sentence related to C-TACE and mention why B-TACE is better.
I think paper can be accepted after Minor revision.
Author Response
Reviewer 2
- 1.Why the ratio of man and women are huge, write the reason.
# Thank you for your valuable comment. Hepatocellular carcinoma has male predominance worldwide (male to female ratio of 2–3:1), and a similar trend is observed in South Korea as well (1, 2). This is likely related to a clustering of risk factors among men as well as differences in sex hormone (1). We strongly agree with your opinion that including this information would increase the comprehensiveness of the content. But unfortunately, due to word count limitation, adding this may necessitate the removal of other content. If considered essential, we can modify other content to accommodate the addition of the above information
- For age, did you follow the medical worldwide rule mention, cite it in table 1.
# We were unable to find specific references for this comment. I referred to the format of tables published in Disease and presented the age in Table 1 as mean ± SD. If further modifications are necessary, please provide more detailed guidance, and we will incorporate them accordingly.
- Expand the abbreviation AST and ALT for more understanding when use for the first time
# In the manuscript we submitted initially, we provided an explanation for the abbreviations of AST and ALT in the first time. Thank you for your comment (Lines 156-157).
- In introduction author should add the names of authors previous work in this field.
# Thank you for your valuable comment. We added the first author in this field and revised the sentence (line 57).
- In Table 2, the words on x-axis are blurred make it clear for one eye.
# We appreciate the reviewer for this query. In the manuscript we submitted, there is no Table 2. If this is in reference to Table 1, we have adjusted the text layout to make it more proper form.
- Conclusion is very short, elaborate it with your best results in discussion.
# Thank you for this helpful comment. We have now added further details to the Conclusion (lines 359-362).
- “in conclusion” in abstract is not meaningful. Use word “we hope”.
# Thank you for helpful suggestions. We have now changed the sentence accordingly (lines 26-27)
- Remove all type mistakes and space issue from the manuscript.
# We apologize for our mistake. We have corrected these in the manuscript.
- There are some complications linked with BTACE, did you check it mention it.
# Thank you for your valuable comment. Complications that can occur after B-TACE are similar to those that can occur after C-TACE, and several studies have been reported that there is no significant difference in the rate of complications (3-6). We have added this point to the Discussion section (lines 326-329).
- Before starting BTACE, give one sentence related to C-TACE and mention why B-TACE is better.
# Thank you for helpful suggestions. We have now added these points to the Introduction section (lines 53-54).
- Llovet JM, Kelley RK, Villanueva A, Singal AG, Pikarsky E, Roayaie S, et al. Hepatocellular Carcinoma. Nature Reviews Disease Primers (2021) 7(1):6. doi: 10.1038/s41572-020-00240-3.
- Kim BH, Park JW. Epidemiology of Liver Cancer in South Korea. Clin Mol Hepatol (2018) 24(1):1-9. Epub 2017/12/19. doi: 10.3350/cmh.2017.0112.
- Kim PH, Gwon DI, Kim JW, Chu HH, Kim JH. The Safety and Efficacy of Balloon-Occluded Transcatheter Arterial Chemoembolization for Hepatocellular Carcinoma Refractory to Conventional Transcatheter Arterial Chemoembolization. Eur Radiol (2020) 30(10):5650-62. doi: 10.1007/s00330-020-06911-9.
- Chu HH, Gwon DI, Kim GH, Kim JH, Ko GY, Shin JH, et al. Balloon-Occluded Transarterial Chemoembolization Versus Conventional Transarterial Chemoembolization for the Treatment of Single Hepatocellular Carcinoma: A Propensity Score Matching Analysis. Eur Radiol (2023) 33(4):2655-64. Epub 20221206. doi: 10.1007/s00330-022-09284-3.
- Golfieri R, Bezzi M, Verset G, Fucilli F, Mosconi C, Cappelli A, et al. Balloon-Occluded Transarterial Chemoembolization: In Which Size Range Does It Perform Best? A Comparison of Its Efficacy Versus Conventional Transarterial Chemoembolization, Using Propensity Score Matching. Liver Cancer (2021) 10(5):522-34. Epub 2021/11/02. doi: 10.1159/000516613.
- Golfieri R, Bezzi M, Verset G, Fucilli F, Mosconi C, Cappelli A, et al. Retrospective European Multicentric Evaluation of Selective Transarterial Chemoembolisation with and without Balloon-Occlusion in Patients with Hepatocellular Carcinoma: A Propensity Score Matched Analysis. Cardiovasc Intervent Radiol (2021) 44(7):1048-59. Epub 20210311. doi: 10.1007/s00270-021-02805-5.

Reviewer 3 Report
Nicely written manuscript on an interesting technique. The retrospective design and the limited sample size are major limitations, addressed as such by the authors in the discussion.
Do the authors have some survival data? If so, could they report the KM curves?
THe authors should comment more on the safety of the technique with its potential impact on the prognosis (on this regard, cite the recent series PMID: 34683182 )
The authors should comment also on the concept of post-recurrence survival after these curative treatments and how to address this issue (cite the study PMID: 25085684)
A table summarizing all the outcomes would be useful
Author Response
Reviewer 3
Nicely written manuscript on an interesting technique. The retrospective design and the limited sample size are major limitations, addressed as such by the authors in the discussion.
# Thank you for your valuable comment. We mentioned these study limitations in the Discussion section.
Do the authors have some survival data? If so, could they report the KM curves?
# We appreciate the reviewer for this valid query. We were unable to obtain survival data because no patient died during the entire follow-up period among the patients included in this study.
The authors should comment more on the safety of the technique with its potential impact on the prognosis (on this regard, cite the recent series PMID: 34683182)
# Thank you for this important comment. We strongly agree that transient increase of AST and ALT level can be the reliable and objective clinical marker for predict treatment response after TACE. So, we have now added these points to the Discussion section with citing the suggested reference.
The authors should comment also on the concept of post-recurrence survival after these curative treatments and how to address this issue (cite the study PMID: 25085684)
# Thank you for this helpful suggestion. We agree with your opinion that the concept of post-recurrence survival and analysis of prognostic factors influencing post-recurrence survival are very important for treatment strategy and management of disease recurrence. However, we were unable to incorporate this into our current work because among 25 patients, only two showed disease progression, and there were no deaths. This is likely due to the favorable treatment effect in cases of single HCC, resulting in no patient deaths during the follow-up period. We are willing to include the analysis of post-recurrence survival after B-TACE in our future studies. Thank you again.
A table summarizing all the outcomes would be useful
# Thank you for this helpful suggestion. We agree with your opinion that summarizing the main results to the table would be beneficial. Therefore, we have now added the summarized outcomes of the study to the new table (lines 261; Table 2).

Round 2
Reviewer 3 Report
The revised version of the manuscript is OK. Thank you!